# Abdominal and Thoracic Imaging Features in Children with MIS-C

Elena Ilieva [1], Vilyana Kostadinova [1], Iren Tzotcheva [2], Nadezhda Rimpova [3], Yordanka Paskaleva [2] and Snezhina Lazova [2,4,*]

1. Department of Diagnostic Imaging, University Emergency Hospital (UMHATEM) "N. I. Pirogov", bul. "General Eduard I. Totleben" 21, 1606 Sofia, Bulgaria
2. Pediatric Department, University Emergency Hospital (UMHATEM) "N. I. Pirogov", bul. "General Eduard I. Totleben" 21, 1606 Sofia, Bulgaria
3. Medical Faculty, Medical University of Sofia, ul. "Zdrave" 2, 1431 Sofia Center, Sofia and University Hospital "Sofiamed", bul. "Jawaharlal Neru" 23, 1336 Sofia, Bulgaria
4. Healthcare Department, Faculty of Public Health, Medical University of Sofia, ul. "Byalo more" 8, 1527 Sofia, Bulgaria
* Correspondence: snejina@lazova.com

**Abstract:** (1) Background: Currently, multisystem inflammatory syndrome in children (MIS-C) is diagnosed based on clinical symptoms and laboratory findings of inflammation in the body. Once MIS-C is diagnosed, children will need to be followed over time. The imaging modalities most commonly used in the evaluation of patients with MIS-C include radiographs, ultrasound (US), and computed tomography (CT). Our study aims to summarise the literature data for the main gastrointestinal and pulmonary imaging features in children diagnosed with MIS-C and to share a single-centre experience. (2) Methods: We present the imaging findings in a cohort of 51 children diagnosed with MIS-C, admitted between December 2020 and February 2022. Imaging studies include chest and abdominal radiographs, thoracic, abdominal, and neck US and echocardiography (ECHO), and CT of the chest, abdomen, and pelvis. (3) Results: In accordance with the results in other studies, our observations show predominantly gastrointestinal involvement (GI) with ascites (33/51, 65%) and lymphadenopathy (19/51, 37%), ileitis or colitis (18/51, 35%), some cases of splenomegaly (9/51, 18%), hepatomegaly (8/51, 16%), and a few cases of renal enlargement (3/51, 6%) and gallbladder fossa oedema/wall thickening (2/51, 4%). Most common among the thoracic findings are posterior–basal consolidations (16/51, 31%), pleural effusion (14/51, 27%), and ground-glass opacities (12/51, 24%). We also register the significant involvement of the cardiovascular system with pericarditis (30/51, 58%), pericardial effusion (16/51, 31%), and myocarditis (6/51, 12%). (4) Conclusions: Radiologists should be aware of those imaging findings in order to take an important and active role not only in applying an accurate diagnosis, but also in the subsequent management of children with MIS-C. Radiological findings are not the primary diagnostic tool, but can assist in the evaluation of the affected systems and guide treatment.

**Keywords:** multisystem inflammatory syndrome in children; Coronavirus Disease 2019; abdominal imaging; chest imaging; ultrasound; computed tomography scan; echocardiography

## 1. Introduction

When the Coronavirus Disease 2019 (COVID-19) pandemic first broke out in 2019, the SARS-CoV-2 infection was perceived as a respiratory condition mainly affecting adults. However, as the spread progressed, clinical reports of extrapulmonary manifestations started accumulating, affecting all age groups. Therefore, special attention was paid to the paediatric age group, where the novel nosological entity multisystem inflammatory syndrome in children (MIS-C), associated with COVID-19, was defined. The case definition

criteria for MIS-C are already described [1] with an estimated incidence rate of 5.1/100 000 [2], primarily affecting children aged 6 to 12 years [3].

The clinical presentation comprises a variety of symptoms. Overlap presentation is undoubtful, but COVID-19 individuals present with respiratory symptoms, whereas rash, vomiting, and diarrhoea are more frequent in MIS-C. The condition is seen by clinicians as a severe complication of SARS-CoV-2 infection. Authors raise awareness not only on differential diagnosis and laboratory findings, but also on imaging features, as they might indicate the correct diagnosis of MIS-C. This article focuses on gastrointestinal and pulmonary imaging findings in presenting paediatric patients with MIS-C. Gastrointestinal symptoms are dominant in patients with MIS-C, reported to be due to infectious lymphadenitis as well as bowel-wall ischaemia secondary to vasculitis [4,5].

Our study aims to summarise the literature data for the main gastrointestinal and pulmonary imaging features and modalities in children diagnosed with MIS-C and to present single-centre observations.

## 2. Materials and Methods

We present the imaging findings in a cohort of 51 children with signs and symptoms meeting the diagnostic criteria for MIS-C and admitted to the University Emergency Hospital in Sofia between December 2020 and February 2022.

The imaging studies undertaken during hospital admission of these patients include chest and abdominal radiographs, thoracic, abdominal, neck, and testicular ultrasound (US) and echocardiography (ECHO), and CT (computed tomography) scans of the chest, abdomen, and pelvis.

The thoracic imaging studies were evaluated for [6,7]:

(1)   Parenchymal lung abnormalities on X-ray (Figure 1) and CT (Figure 2), including ground-glass opacity, posterior basal consolidation, subsegmental atelectasis, air bronchogram, and crazy paving.
(2)   Pleural abnormalities including pleural effusion (Figures 1 and 2) and pneumothorax.
(3)   Cervical, mediastinal, and axillary lymphadenopathy.
(4)   Cardiovascular abnormalities including cardiomegaly (Figure 1c) and signs of pericarditis, myocarditis, and pericardial effusion.

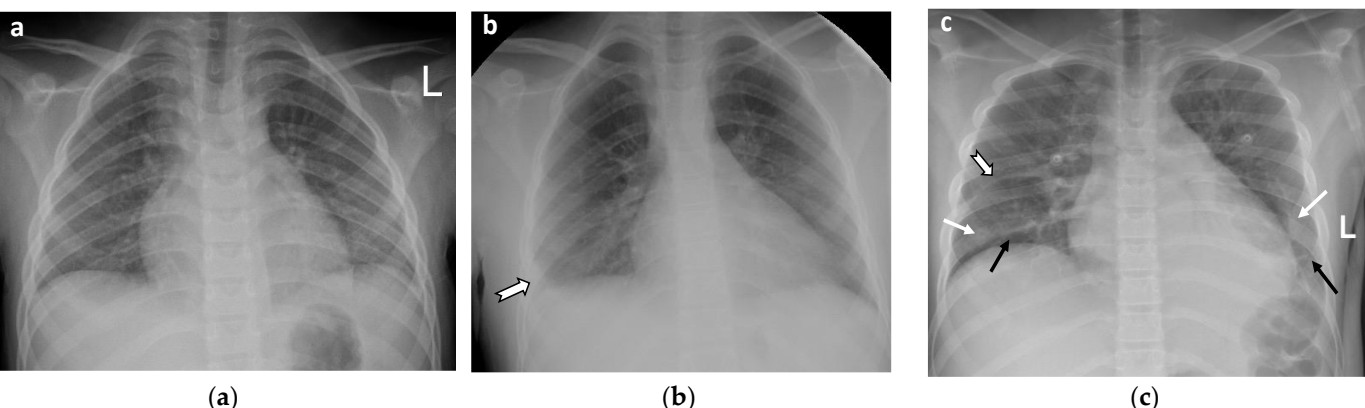

(a)                                             (b)                                             (c)

**Figure 1.** X-rays of different paediatric patients with multisystem inflammatory syndrome in children (MIS-C). (**a**) AP X-ray in an 8-year-old boy with MIS-C without specific lung findings. (**b**) AP X-ray in a 6-year-old girl demonstrates right sided pleural effusion (thick arrow). (**c**) AP X-ray in a 13-year-old boy demonstrates peripheral ground-glass opacities (white arrow) with interstitial thickenings in lower lung lobes (black arrow), small right-sided interlobar effusion (thick arrow), and an enlarged cardiac silhouette.

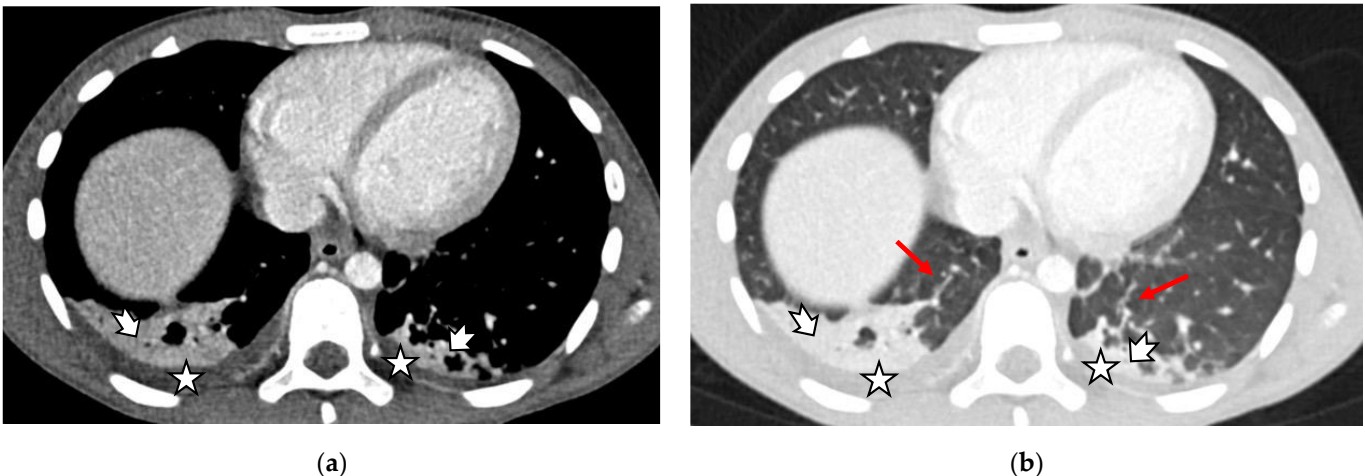

(**a**) (**b**)

**Figure 2.** Axial contrast-enhanced chest CT (CECT) scans—abdominal (**a**) and lung (**b**) window showing bilateral small pleural effusions (white star), posterobasal areas of consolidation (thick white arrow), and interstitial thickening (red arrows).

The abdominal imaging studies were evaluated for [6,7]:

(1)    Size abnormalities of the liver and spleen (Figure 3), kidneys, and pancreas.
(2)    Parenchymal abnormalities of the solid organs (heterogeneous echotexture of the liver or spleen, increased echogenicity of the renal parenchyma compared to the liver, and abnormal or heterogeneous attenuation).
(3)    Abnormalities of the hollow visceral organs, including distension and wall thickening of the gallbladder (Figure 4), stomach, bowel, small bowel (Figure 5A, and urinary bladder (on CT and ultrasound). They were considered thickened if >3 mm. Additionally, the cross-sectional diameter of appendix is considered enlarged if >6 mm (Figure 5B).
(4)    Peritoneal abnormalities—ascites (Figure 6), fluid collections, and pneumoperitoneum.
(5)    Mesenteric and retroperitoneal lymphadenopathy (Figure 7) with mesenteric lymph nodes considered enlarged if >5 mm in short axis and retroperitoneal lymph nodes considered enlarged if >9 mm in short axis.

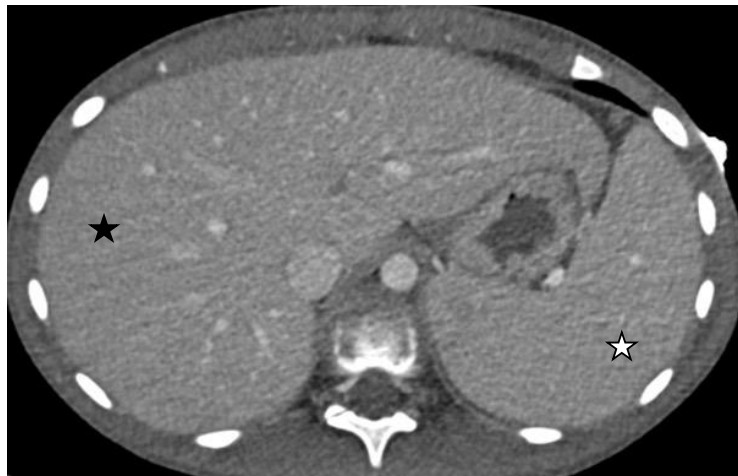

**Figure 3.** Axial contrast-enhanced abdominal CT scan demonstrates liver (black star) and splenic (white star) enlargement.

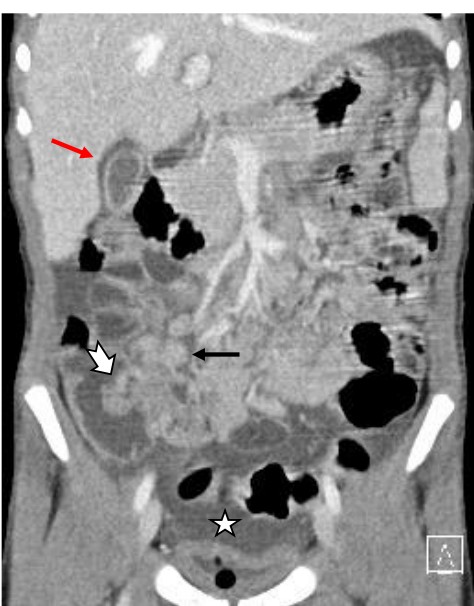

**Figure 4.** Coronal reconstruction of a contrast-enhanced abdominal CT scan. Image shows enlarged ileocecal lymph nodes (black arrow), ascites (white star), thickened gall bladder wall (red arrow), and thickened terminal ileum walls with submucosal enhancement (thick white arrow).

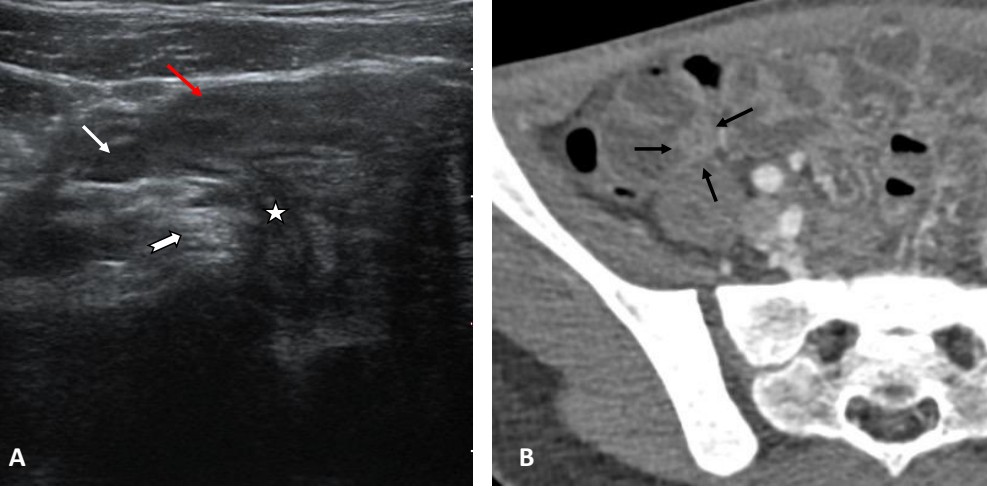

**Figure 5.** Iliac fossa changes. (**A**) Abdominal ultrasound image obtained with linear probe demonstrates thickened, hypoechoic terminal ileum wall (red arrow), slightly enlarged ileocecal lymph node (white arrow), hyperechoic iliac fossa fat due to oedema (thick white arrow), and normal-sized appendix (white star). (**B**) Axial contrast-enhanced chest CT scan shows bended normal-sized appendix (black arrows).

For each patient and each modality, a finding was considered present if the finding was identified in any single imaging examination. For example, if a patient had multiple abdominal tests (X-ray, ultrasound, CT) and a single one demonstrated enlarged lymph nodes, this patient was recorded as having enlarged lymph nodes on all of the imaging studies.

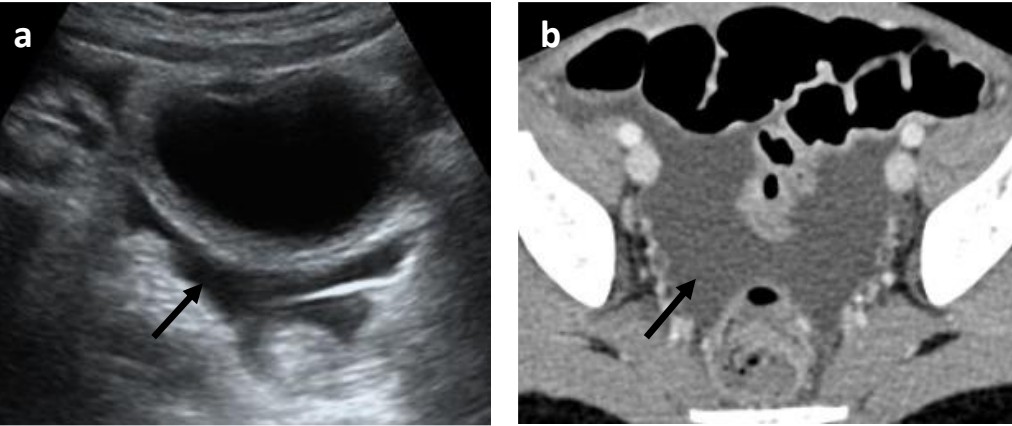

**Figure 6.** Ascites (black arrows) on ultrasound image obtained with curvilinear transducer (**a**) and on axial CECT image (**b**).

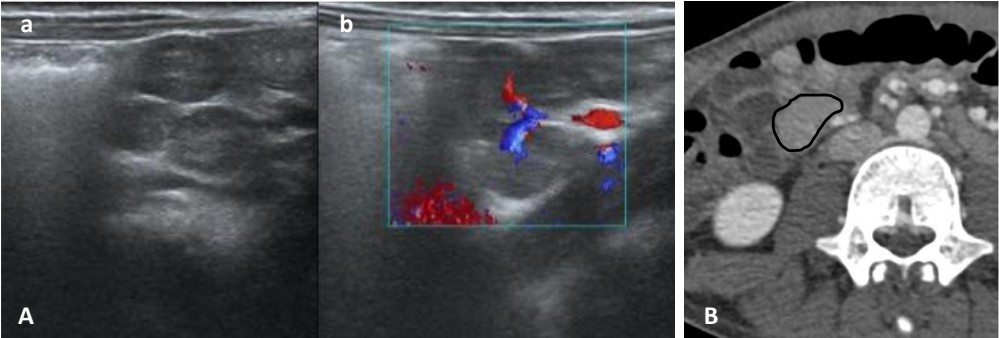

**Figure 7.** Enlarged iliac fossa lymph nodes. (**A**) Abdominal ultrasound images obtained with linear probe demonstrate enlarged ileocecal lymph nodes (**a**) with hilar hypervascularity at colour Doppler ultrasound (**b**). (**B**) Axial contrast-enhanced abdominal CT scan of right iliac fossa shows confluent mesenteric lymph nodes (black contour).

All imaging examinations were performed and analysed by certified roentgenologists specialised in paediatric pathology. The presented results are exclusively observational and descriptive without statistical data analysis. None of the included images are personalised.

The data included in the current paper were collected as part of a bigger real-life study approved by the Ethics Committee (protocol № 123-20/23.12.2020) of the University Hospital "N.I. Pirogov" and the Helsinki Declaration. All parents signed informed consent before including their children in the study.

## 3. Results

### 3.1. Demography

The most prominent clinical symptoms at admission were abdominal pain and fever. Several of these children were directed for hospital admission with a working diagnosis of acute appendicitis. The demographic characteristics of the included children and a list of the performed imaging modalities are presented in Table 1.

Seven children underwent surgical intervention—five laparotomies with appendectomy and two thoracenteses for treating pleural effusion. All cases finished with recovery without lethality.

**Table 1.** Main demographic characteristics of the cohort and list of the performed imaging modalities.

| Characteristics | | Number | Percent |
|---|---|---|---|
| Number | | 51 | |
| Sex, %male | | 37 | 72.5% |
| Mean age, years | | 8.72 (1–16) | |
| Digestive symptoms | Vomiting | 26 | 51% |
| | Diarrhoea | 23 | 45% |
| | Any | 37 | 72.5% |
| Respiratory symptoms<br>Cough, chest pain, different degrees of respiratory distress | | 23 | 45% |
| Heart involvement | Any (clinical, laboratory, imaging, and functional evidence) | 26 | 51% |
| | Myocarditis | 6 | 12% |
| | Myocarditis rapidly progressing to cardiovascular shock | 3 | 5.9% |
| Renal involvement (severe acute renal failure) | | 1 | 2% |
| **Performed Imaging Tests** | | **Number** | **Percent** |
| Chest X-ray | | 36 | 70.5% |
| Abdominal X-ray | | 10 | 19.6% |
| Thoracic CT | | 22 | 43.1% |
| Abdominal CT | | 20 | 39.2% |
| Thoracic and abdominal CT | | 19 | 37.2% |
| Abdominal US | | 51 | 100% |
| Thoracic US | | 35 | 68.6% |
| Echocardiography | | 37 | 72.5% |
| Neck US | | 5 | 9.8% |
| Testicular US | | 3 | 5.9% |

### 3.2. Main Pulmonary Imaging Findings

The main thoracic imaging findings are summarised in Table 2, followed by representative images of chest X-ray and axial contrast-enhanced chest CT (CECT) scans (Figures 1 and 2).

**Table 2.** Main thoracic imaging findings—chest CT scan, chest X-ray, thoracic US, and echocardiography.

| Imaging Test, N Tested | Pathological Findings | Number | % Tested | % All |
|---|---|---|---|---|
| | CT ground glass | 12 | 54.5% | 23.5% |
| | Crazy paving pattern | 1 | 4.5% | 2.0% |
| | Subsegmental atelectasis | 5 | 22.7% | 9.8% |
| | Posterior basal consolidation | 16 | 72.7% | 31.4% |
| | Air bronchogram | 3 | 13.6% | 5.9% |
| Chest CT scan findings, N 22 | Pleural effusion | 14 | 63.6% | 27.5% |
| | Pericardial effusion | 4 | 18.2% | 7.8% |
| | Cervical lymphadenitis | 10 | 45.5% | 19.6% |
| | Mediastinal lymphadenitis | 4 | 18.2% | 7.8% |
| | Axillar lymphadenitis | 3 | 13.6% | 5.9% |
| | Thymic enhancement | 5 | 22.7% | 9.8% |
| Chest X-ray findings, N 36 | Lung and/or heart pathology | 16 | 44.4% | 31.4% |
| | Pleural effusion | 30 | 86% | 58.82% |
| | Small | 24 | 69% | 47.06% |
| Thoracic US, N 35 | Moderate | 4 | 11% | 7.84% |
| | Large | 1 | 3% | 1.96% |
| | Fibrinous | 1 | 3% | 1.96% |
| | Bilateral | 20 | 57% | 39.22% |
| Echocardiography, N 37 | Pericarditis | 29 | 78% | 56.86% |
| | Small pericardial effusion | 16 | 43% | 31.37% |

### 3.3. Main Gastrointestinal Imaging Findings

The main gastrointestinal/abdominal imaging findings are summarised in Table 3. Examples of the main imaging abnormalities are presented in Figures 3–7.

**Table 3.** Main abdominal imaging findings—abdominal CT, abdominal X-ray, abdominal US.

| Imaging Test, N Tested | Pathological Findings | Number | % Tested | % All |
|---|---|---|---|---|
| | Mesenteric lymphadenitis | 14 | 70% | 27.5% |
| | Colitis | 5 | 25% | 9.8% |
| | Terminal ileitis | 10 | 50% | 19.6% |
| | Terminal ileitis and appendicitis | 4 | 20% | 7.8% |
| | Hepatic enlargement | 8 | 40% | 15.7% |
| | Splenic enlargement | 9 | 45% | 17.6% |
| Abdominal CT scan, N 20 | Renal pathology, any | 5 | 25% | 9.8% |
| | Renal enlargement | 3 | 15% | 5.9% |
| | Pyelectasis | 1 | 5% | 2.0% |
| | Nephrocalcinosis | 1 | 5% | 2.0% |
| | Pancreatic oedema | 1 | 5% | 2.0% |
| | Gallbladder wall thickening | 2 | 10% | 3.9% |
| | Ascites | 17 | 85% | 33.3% |
| | Enteric gas | 3 | 30% | 5.9% |
| Abdominal X-ray, N 10 | Hydroaeric | 1 | 10% | 2% |
| | Enteric gas plus hydroaeric | 5 | 50% | 9.8% |
| | Pneumoperitoneum | 1 | 1% | 2.0% |

**Table 3.** *Cont.*

| Imaging Test, N Tested | Pathological Findings | Number | % Tested | % All |
|---|---|---|---|---|
| | Gallbladder sludge | 2 | 3.9% | 3.9% |
| | Gallbladder wall thickening | 2 | 3.9% | 3.9% |
| | Gallbladder concernments | 1 | 2.0% | 2% |
| | Mesenteric lymphadenitis | 28 | 54.9% | 54.9% |
| | Mesenteric lymphadenitis Diffuse | 9 | 17.6% | 17.6% |
| | Mesenteric lymphadenitis Ileocecal | 7 | 13.7% | 13.7% |
| Abdominal US, N 51 | Mesenteric lymphadenitis Diffuse and ileocecal | 4 | 7.8% | 7.8% |
| | Enteritis/enterocolitis | 4 | 7.8% | 7.8% |
| | Gas/liquid intestinal | 7 | 13.7% | 13.7% |
| | Ascites | 33 | 64.7% | 64.7% |
| | Hepatic enlargement | 5 | 9.8% | 9.8% |
| | Splenic enlargements | 5 | 9.8% | 9.8% |
| | Renal pathology, any | 5 | 9.8% | 9.8% |
| | Renal pathology, enlarged | 2 | 3.9% | 3.9% |
| | Renal pathology, hyperechogenic | 1 | 2% | 2% |
| | Renal pathology, hydronephrosis | 1 | 2% | 2% |
| | Renal pathology, hydrocalicosis | 1 | 2% | 2% |

### 3.4. Other Imaging Modalities and Findings

Other imaging studies were performed on clinical indications such as neck and testicular swelling, as listed in Table 4. According to the US examination, the testicular swelling was diagnosed as unilateral hydrocele. In 40% of the children with neck swelling, the US data showed lymphadenitis and parotitis; in 60%, only lymphadenopathy was found without parotid gland involvement. Additionally, thymic changes with profound enhancement on contrast enhanced CT (CECT) along with slight or no enlargement of the gland were noted in some of our patients (6/51, 12%), usually the ones with more severe pulmonary involvement and severe lymphopenia (26% of all patients that underwent thoracic CT) (Figure 8).

**Table 4.** Other imaging findings—neck and testicular US.

| Imaging Test, N Tested | Pathological Findings | Number | % Tested | % All |
|---|---|---|---|---|
| Neck US, N 5 | Lymphadenopathy | 3 | 60% | 5.88% |
| | Lymphadenitis and parotitis | 2 | 40% | 3.92% |
| Testicular US, N 3 | Unilateral hydrocele | 3 | 100% | 5.88% |

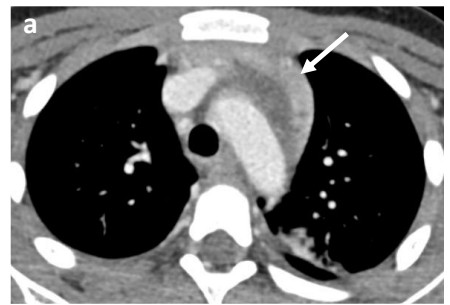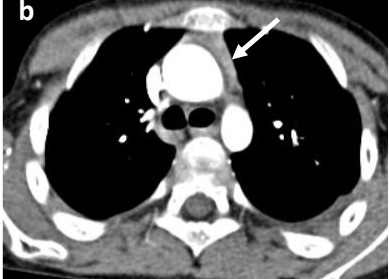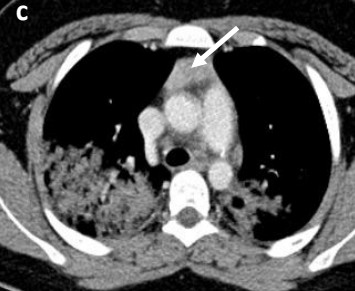

**Figure 8.** Axial CECT scans showing profound contrast enhancement and slight enlargement of thymic gland in a 6-year-old boy (**a**), 7-year-old boy (**b**), and 13-year-old boy (**c**).

## 4. Discussion

Currently, multisystem inflammatory syndrome in children (MIS-C), a potentially serious illness that appears to be a delayed, post-infectious complication of COVID-19 infection, is diagnosed based on clinical symptoms and laboratory findings of inflammation in the body. Once MIS-C is diagnosed, children will need to be followed over time [8]. According to several studies that have examined and evaluated the imaging findings of MIS-C, imaging is important for the evaluation of abdominal and thoracic manifestations of the disease in the follow-up of hospitalised patients and plays a key role in the differential diagnosis. It is now established that the differential diagnosis of abdominal pain is a serious diagnostic challenge, after including COVID-19 infection and MIS-C in consideration, alongside with appendicitis, mesenteric adenitis, and other less common causes of abdominal pain [9]. Proper utilisation of different imaging modalities and interpretation of the key imaging findings are essential for effective patient management and treatment.

The imaging modalities that are most commonly used for the evaluation of patients with MIS-C include chest and abdominal radiographs, abdominal ultrasound and echocardiography, and CT of the chest, abdomen, and pelvis.

Ultrasonography remains the main imaging modality for children with abdominal pain, but is not the only approach. This method avoids the exposure to ionising radiation, is safe, simple (after adequate training), and can be used at the bedside for detecting many pathological findings. Although respiratory signs are generally not part of the MIS-C presentation, and therefore found in a smaller percentage of children, there are a few articles about the advantages and benefits of lung US. Musolino et al. suggest performing point-of-care lung ultrasound in febrile patients with high levels of inflammatory indices and clinical suspicion of MIS-C, or without a certain diagnosis during the ongoing COVID-19 pandemic [10]. The findings of bilateral diffuse B-lines associated with pleural irregularities and pleural effusions may support the diagnosis of a systemic inflammatory disease and may suggest MIS-C, even in the absence of respiratory symptoms. Patients that were investigated in Musolino's article underwent US evaluation in a sitting position and 10 areas (two anterior, two posterior, and one axillary area for each hemithorax) were scanned [10]. The US findings considered were: pleural effusion, pleural irregularities (including subpleural consolidations), parenchymal consolidations, and B-lines. B-line density was defined by the finding of multiple B-lines, while white lung density was defined as increased lung echogenicity with the disappearance of normal A-lines [10]. The main characteristic observed in our study is the different degrees of pleural effusion with or without subpleural consolidations and atelectatic changes in the basal lung parenchyma. In most of the children, the focus of the thoracic US study was the pleura rather than the lung parenchyma. Thus, the A- and B-lines are not analysed.

Chest radiography, called chest X-ray (CXR), is in use everywhere worldwide and involves a 30–70 times lower dose of radiation than a computed tomography scan (CT). It is usually used as an initial investigation in COVID-19 despite its low sensitivity and specificity. In MIS-C, it is a modality of choice mainly because of the presented cardiovascular abnormalities such as peri-bronchial cuffing, perihilar interstitial thickening, and heart enlargement [6]. In our cohort, we performed screening chest X-rays in 36 children (70.5%). Pathological findings were observed in 44.4% of the examined children. The typical chest X-ray findings were unilateral pleural effusion, lower lung lobe interstitial thickening, peripheral ground-glass opacities, and enlarged cardiac silhouette [11]. Meanwhile, CT is fast, but expensive, with a higher radiation dose and capacity still lacking in many countries; however, it is a modality that gives detailed information for the interstitial or alveolar involvement of lung parenchyma (ground-glass opacities, consolidation, atelectasis, mosaic pattern, and reverse halo sign), and also for the distribution (unilateral or bilateral, central, or peripheral), zonal distribution (upper, middle, or upper zone), and focality (single, multifocal or diffuse) of the identified parenchymal pathologies. Other pathologies such as small pleural effusions and mediastinal and hilar lymphadenopathy are also presented with CT [11]. According to the literature data, in children with MIS-C associated with

COVID-19, the most commonly described thoracic features include ground-glass opacities (9%), pulmonary consolidations (39%), pleural effusions—the majority of which are bilateral (39 and 63%)—as well as cardiomegaly, pancarditis (43%), congestive heart failure, or pulmonary oedemas [12]. We also observed typical lung involvement for acute COVID-19 and MIS-C in the children indicated for chest CT. Furthering previous studies, the most common among the thoracic findings in our cohort were pleural effusion (14/22, 63.6%), posterior–basal consolidations (12/22, 72.7%) and ground-glass opacities (12/22, 54.5%). In only 4.5% (1/22) of the performed chest CT studies, we observed crazy paving patterns.

The mediastinal and axillar lymphadenopathy or both observed in chest CT scan is observed by other authors as well [6,13]. In our cohort, cervical lymphadenomegaly is detected on the chest CT scan in 45.5% of the examined children (10/22), mediastinal in 18.2% (4/22), and axillar in 13.6% (3/22). This finding is suggested to be a result of the hyperinflammatory state characterising the MIS-C in contrast with the acute COVID-19 in children [6].

Thanks to its unique ability to directly image myocardial necrosis, fibrosis, and oedema, cardiac magnetic resonance (CMR) is now considered the first tool for the non-invasive assessment of patients with suspected myocarditis—an abnormality demonstrated in many studies [14]. Up to 80% of children with MIS-C may have cardiac involvement on a spectrum of severity. Other cardiac manifestations include coronary artery aneurysms, conduction abnormalities, and arrhythmias [15]. In our study, we did not include CMR data due to the limited number of tested patients. Using the ECHO and chest CT data, we also register the significant involvement of the cardiovascular system with pericarditis (30/51, 58%), pericardial effusion (20/51, 39%), and myocarditis (6/51, 12%). The cardiac studies were performed by paediatric cardiologists and the results are mentioned briefly here, as this is not the focus of this article.

Numerous articles and reviews have been published reporting gastrointestinal disorders to be among the most frequent symptoms in children with MIS-C. Cheap, available, and safe ultrasound is the perfect first-line modality in every hospital [16,17]. It is very important in establishing lymph node short-axis diameter enlargement greater than 8 mm, or bowel-wall thickening greater than 2 mm, or confirming or rejecting the differential diagnosis of appendicitis, intussusception, or mesenteric lymphadenitis. However, ultrasound is less sensitive (55% vs. 98%) and specific (85% vs. 100%) than CT in diagnosing appendicitis [18]. In a case series of four children, the appendix was not satisfactorily depicted with US. They underwent CT, which successfully demonstrated no evidence of appendicitis while confirming the presence of MIS-C-like features in all cases [19].

Mesenteric lymphadenitis, seen in patients with MIS-C, is the main cause of severe abdominal pain. Therefore, US images in MIS-C are rarely normal. Persistent US findings are bowel-wall thickening involving the right iliac fossa (21%), the terminal ileum and/or cecum (47%), and lymphadenomegaly (47%). Increased periportal echogenicity, pericholecystic oedema, and mild gallbladder-wall thickening and sludge were also reported [11]. Some authors report hepatosplenomegaly at a rate of 28% to 34% [20]. In our cohort, using the US method, enlarged liver and spleen was observed in 9.8%. Other rarer pathological findings were renal enlargement, renal hyper echogenicity, and hydronephrosis.

There are numerous reports of severe acute abdominal pain mimicking acute appendicitis, including some studies where patients received surgery, but only revealed mesenteric lymphadenitis [21,22]. CT of the abdomen and pelvis is a modality of choice, especially when US is not diagnostic enough to exclude the diagnosis of acute appendicitis, but also intussusception or mesenterial lymphadenitis. It is generally performed by radiologists, as US findings depict non-specific inflammation. The general indication for abdominal CT scan in our cohort was the severe abdominal pain and suspected acute appendicitis. CT findings of acute appendicitis are usually one or more abnormalities from these: dilated appendix with a diameter of more than 6 mm, wall thickening of the appendix or terminal ileum more than 2 mm, adjacent mesenteric fat stranding, mesenteric lymph nodes, appendicolith, and peri-intestinal fluid [23]. In children with MIS-C, the

most common findings include enlarged lymph nodes and free intraabdominal fluid, but normal appendix with no fat stranding around it. The most common CT and MRI findings in MIS-C patients include ascites (71%), intestinal/colonic inflammation (57%), and mesenteric adenopathy [24].

The most common abdominal abnormalities observed in MIS-C are extensive right iliac fossa inflammatory changes, presented with ascites (38% and 53%), mesenteric lymphadenopathy (13% and 47%), bowel-wall thickening (19% and 21%) and normal appendix. Hepatosplenomegaly, echogenic kidneys, and gallbladder- and bladder-wall thickening are also among the most commonly encountered abnormalities on abdominal images [6,7,11,12,25].

Similar to the results in other studies for imaging findings in MIS-C, our observations using a combination of CT and US data show predominantly gastrointestinal involvement with ascites (35/51, 69%) and lymphadenopathy (26/51, 51%), ileitis or colitis (18/51, 35%), some cases of splenomegaly (12/51, 24%), hepatomegaly (11/51, 22%), and a few cases of renal enlargement (3/51, 6%) and gallbladder fossa oedema/wall thickening (4/51, 8%).

Limited data exist in the literature regarding reactive thymic hyperplasia in patients with COVID-19 acute respiratory distress syndrome [26]. However, to our knowledge, the findings observed in our cohort were not mentioned in previous studies associated with MIS-C. A study highlighting thymic involvement in adults concluded that thymic reactivation and enlargement in COVID-19 patients reflect the triggering of compensatory mechanisms, which the authors find to be a good prognostic sign, related to low mortality. On the other hand, lack of thymic activity contributes to a worse prognosis [26]. The authors of another study on adult CT findings in COVID-19 refer to thymic hyperplasia as the "only positive finding" and advise radiologists to actively look for it [27]. However, thymic enlargement in adults is unusual. It can only be seen in some pathologic conditions, whereas in children, the thymus is a major organ in T-cell maturation and production until its natural involution after puberty onset. Thus, thymic persistence on CT and X-ray images is a common finding in paediatric patients. The thymus-modulated adaptive immune response, mainly due to regulatory T-cell production in early life, is proposed as one of the major protective factors against a more severe COVID-19 infection. Numerous literature reports confirm paediatric COVID-19 to be significantly less severe than adult COVID-19. Treatment strategies to prevent thymic atrophy and stimulate thymic function are suggested [28]. We confirm the results of other studies, concluding that thymic enlargement is more likely to be seen in patients with a more severe pulmonary involvement mainly due to an efficient immune response, since those patients in our study exhibited full recovery. To better clarify the thymic role in COVID-19 infection in children, further studies, including cytokine and T-lymphocyte plasma activity during infection, should be evaluated.

## 5. Practical Implications and Limitations

An awareness of imaging findings in the setting of COVID-19 and its expected multi-organ involvement will aid radiologists in the rapid and accurate assessment of these complex cases. This, on the other hand, will help in the clinical improvement of patients, providing appropriate management and treatment after an accurate diagnosis [7,11]. Further retrospective and prospective studies will be required to provide evidence of thymic involvement as well as the involved molecular and cellular mechanisms in severe cases of MIS-C [26]. The main limitation of our real-life single-centre study is the small number of MIS-C cases and the exclusively descriptive data.

## 6. Conclusions

Imaging studies play an important role in conclusions in diagnostic dilemmas, the evaluation of the spectrum of multisystemic involvement in MIS-C, and in the follow-up of hospitalised patients with MIS-C. Radiological findings are not the primary diagnostic tool, but can assist in the evaluation of the affected systems and guide treatment. In accordance with the results in other studies, our observations show predominantly gastrointestinal

involvement with ascites and lymphadenopathy, ileitis, or colitis. The most commonly observed thoracic findings are posterior–basal consolidations, pleural effusion, and ground-glass opacities. Additionally, thymic changes with profound enhancement on CECT were noted in some cases. The suggested connection with severe pulmonary involvement and severe lymphopenia could be a subject of further studies.

Radiologists and clinicians should be aware of these imaging findings in order to take an important and active role not only in applying an accurate diagnosis, but also in the subsequent management of children with MIS-C.

**Author Contributions:** Conceptualization, S.L. and E.I.; methodology, S.L. and E.I.; software, S.L., E.I. and V.K.; validation, E.I., S.L. and V.K.; formal analysis, S.L. and E.I.; investigation, S.L., I.T. and E.I.; resources, S.L, I.T. and E.I.; data curation, E.I., S.L. and V.K.; writing—original draft preparation, E.I., V.K., N.R., S.L. and Y.P.; writing—review and editing, S.L., E.I., V.K., N.R., Y.P. and I.T.; visualization, E.I., S.L. and V.K.; supervision, S.L.; project administration, S.L. and I.T. All authors have read and agreed to the published version of the manuscript.

**Funding:** This research received no external funding.

**Institutional Review Board Statement:** The study was conducted following the Ethics Committee approved protocol (№123-20/23.12.2020) of the University Hospital "N.I. Pirogov" and the Helsinki Declaration.

**Informed Consent Statement:** All parents signed informed consent to include their children in the study. All children older than 12 years signed additional informed consent on their own.

**Data Availability Statement:** The data presented in this study are available on request from the corresponding author. The data are not publicly available due to restrictions, e.g., privacy or ethics.

**Acknowledgments:** The paediatric surgeons, paediatric anaesthesiologist, and intensive care specialists and the paediatric yard team of the UMHATEM "N.I. Pirogov".

**Conflicts of Interest:** The authors declare no conflict of interest.

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
