# Peer review of "Abdominal and Thoracic Imaging Features in Children with MIS-C"

_gastroent, doi:10.3390/gastroent13040032_

Round 1

Reviewer 1 Report

The manuscript "Abdominal and thoracic imaging features in children with MIS-C" by Ilieva et al. aimed to present the imaging features of MIS-C as an essential element in the diagnosis of this recent pathology with severe implications in children after COVID-19. Even though their aim is important and some aspects presented are necessary for the diagnosis, the manuscript presents essential issues that request a significant revision. The manuscript should be rewritten and better organized as a research article. The authors should choose to have either a narrative review, systematic review, or research article based on their experience.

Some recommendations if they would like to have it as a research paper:

- the abstract: there is a repetition of the imaging studies used in their research;

- keywords: there should be no abbreviated words here;

- the name of the virus should be written correctly: SARS-CoV-2;

- the introduction should be reorganized with few data on the imaging studies used in MIS-C; some information should not be repeated, and some are already well-known;

- there should be a Material and methods section and not present the results directly;

- the Discussion section should not include the results again but analyze the results of this study compared to other studies and present the importance of these results;

- the Conclusions should be based on the results of this study. Therefore, there should not be another repetition of the results.

Author Response

Dear Reviewer,

Thank you for your time in evaluating our manuscript " Abdominal and thoracic imaging features in children with MIS-C" authors: Elena Ilieva, Vilyana Kostadinova, Iren Tzotcheva, Nadezhda Rimpova, Yordanka Paskaleva and Snezhina Lazova, submitted to the journal “Gastroenterology insights”.

The manuscript "Abdominal and thoracic imaging features in children with MIS-C" by Ilieva et al. aimed to present the imaging features of MIS-C as an essential element in the diagnosis of this recent pathology with severe implications in children after COVID-19.

  • Thank you very much for the good overall evaluation of our topic choice.

Even though their aim is important and some aspects presented are necessary for the diagnosis, the manuscript presents essential issues that request a significant revision.

  • Thank you for the valuable comment. We acknowledge that our manuscript may have some issues in conformity with the following comments. We have addressed all the problematic points, and the revisions are marked with the track changes option in the text.

The manuscript should be rewritten and better organized as a research article. The authors should choose to have either a narrative review, systematic review, or research article based on their experience.

  • The referee is right to point out the confusing article structure. We completely agreed, and after receiving an Editor’s permission, we made a general reorganization of the text and changed the article’s type from narrative review to research article based on our
  • In this respect, we made the following changes:
    • The introduction is shortened and focused on imaging studies in MIS-C; the repeated and already well-known information was excluded from the text following the referee's comments and recommendations.
    • We added a Material and methods section following the research article structure. Ethical statements were also included. A copy of the Ethics Committee’s protocol is provided.
    • The result section, including four subsections, is reorganized, following the changed article format with a separate demography subsection. All tables and figures are moved to the belonging subsection. In subsection 3.2. Main pulmonary imaging findings, in table 2, a new row was added - Chest X-ray findings, following the results logic.
    • The discussion section is generally reconstructed and significantly changed. In the discussion revision, we considered all of the referee's comments. The repetitions were removed, and we focused the discussion on the results of this study and the literature data.
    • A new element in the fifth section was added - Practical implications and limitations.
    • The conclusion section is also revised, following the referee's comments and recommendations.

Some recommendations if they would like to have it as a research paper:

  • Thank you for the valuable comments and recommendations.

- the abstract: there is a repetition of the imaging studies used in their research;

  • Thank you for your comment. We revised the abstract, and all repetitions were removed. The abstract’s structure was adapted to the changed article type.

- keywords: there should be no abbreviated words here;

  • Thank you very much for the valuable note. We removed all abbreviations in the keywords.

- the name of the virus should be written correctly: SARS-CoV-2;

  • Thank you for noticing this technical mistake. The name of the virus was corrected in the text;

- the introduction should be reorganized with few data on the imaging studies used in MIS-C; some information should not be repeated, and some are already well-known;

  • We completely agree with the referee for this constructive remark. The introduction is reorganized, shortened and focused on imaging studies in MIS-C; the repeated and already well-known information was excluded from the text following the referee's comments and recommendations.

- there should be a Material and methods section and not present the results directly;

  • Thank you for your comment. The material and methods section was added following the research article structure. Ethical statements were also included. A copy of the Ethics Committee’s protocol is provided.

- the Discussion section should not include the results again but analyze the results of this study compared to other studies and present the importance of these results;

  • The discussion section is generally reconstructed and significantly changed. In the discussion revision, we considered all of the referee's comments. The repetitions were removed, and we focused the discussion on the results of this study and the literature data.

- the Conclusions should be based on the results of this study. Therefore, there should not be another repetition of the results.

  • The conclusion section is revised following the referee's comments and recommendations.

Reviewer 2 Report

The authors focused on the abdominal and thoracic imaging features of these patients in their review of MISC, which is a current issue, and also shared the characteristics of their own cases. The authors wrote the review well. They included discussion with current references. The presentation of demonstrative imaging recordings also enriched the review. I think that this review will contribute to the literature.

The typo on line 52 needs to be corrected (71%.).

I think it is appropriate to publish this work as it is.

Author Response

Dear Reviewer,

Thank you for your time in evaluating our manuscript " Abdominal and thoracic imaging features in children with MIS-C" authors: Elena Ilieva, Vilyana Kostadinova, Iren Tzotcheva, Nadezhda Rimpova, Yordanka Paskaleva and Snezhina Lazova, submitted to the journal “Gastroenterology insights”.

The authors focused on the abdominal and thoracic imaging features of these patients in their review of MISC, which is a current issue, and also shared the characteristics of their own cases. The authors wrote the review well. They included discussion with current references. The presentation of demonstrative imaging recordings also enriched the review. I think that this review will contribute to the literature.

  • Thank you very much for the good overall evaluation of our work.

The typo on line 52 needs to be corrected (71%.).

  • Thank you very much for your comment. We made corrections to this technical mistake.

I think it is appropriate to publish this work as it is.

  • Thank you again for the overall evaluation of our paper as good and reaching the standards.

Reviewer 3 Report

This research dealt with MIS-C radiological features, which is an interesting topic and matches the interest of the journal audience. Even if the study design did not include a control group the study is well written. I suggest improving the reference list.

Author Response

Dear Reviewer,

Thank you for your time in evaluating our manuscript " Abdominal and thoracic imaging features in children with MIS-C" authors: Elena Ilieva, Vilyana Kostadinova, Iren Tzotcheva, Nadezhda Rimpova, Yordanka Paskaleva and Snezhina Lazova, submitted to the journal “Gastroenterology insights”.

This research dealt with MIS-C radiological features, which is an interesting topic and matches the interest of the journal audience.

  • Thank you very much for the good overall evaluation of our work.

Even if the study design did not include a control group the study is well written.

  • Thank you very much for your comment. Our findings are exclusively observational and did not include a control group. In future analysis and study, children with acute COVID-19 could serve as a control group.

I suggest improving the reference list.

  • We completely agree with the referee's comment. The reference list is revised.

Round 2

Reviewer 1 Report

The authors changed their manuscript's style and presented it as a research article based on their experience in imaging in MIS-c patients. This is more appropriate for the data presented. The paper now respects the structure of this kind of paper. The abstract and the sections of the manuscript were improved. All comments were appropriately addressed, and the manuscript may be of genuine interest to readers in this version. Still, there are some editing issues, but these may be corrected through the editing steps after review.